# Auditable Early Stopping for Agentic Routing: Ledger-Verified Run-Wise Certificates under Local DP

## Abstract

We study early stopping for best-first routing in tool-use agents under local differential privacy (LDP), with an auditable, validator-replayable ledger. Our key idea is a *run-wise certificate*: we couple each node's key to the *same* exponential race that realizes leaf perturbations, so the standard halting rule—stop when $\max_{v \in \mathcal{F}} \mathrm{Key}(v) \leq B^*$, where $B^*$ is the incumbent realized leaf value—soundly certifies the realized run. We provide two certified modes on context-indexed prefix–DAGs whose children partition the leaf set. *Exact* mode (known counts) implements lazy offset propagation with winner reuse; *Surrogate* mode (upper bounds only) disables winner reuse and uses a parent-anchored surrogate race; keys are conservative online and can be tightened ex post by a validator via $\kappa = \log(N/N_{\mathrm{ub}})$. A small compiler enforces the partition property, and an admissible, race-independent $M_\tau$ keeps keys sound. A replayable ledger records uniforms, counts, and tie handling; privacy follows from post-processing. Experiments on synthetic graphs and a small real tool-use pipeline show tight stopping, deterministic replay, and low overhead.

## 1 One-Minute Story (Running Example)

You route a question through a tool pipeline (retrieval $\rightarrow$ summarization $\rightarrow$ calculator) under **local DP**: inputs are privatized upstream; routing must be auditable yet privacy-preserving. Your grammar yields a *shared-node* DAG (subtools reused across contexts). Our compiler *context-indexes* nodes to form a *prefix–DAG* whose children *partition* descendant leaves. The router runs *Lazy–A* PaM* with realized path score

$$s_{\mathrm{det}}(P) = -C(P) \qquad \text{(certified deterministic score; PaM/Gumbel may be used as a heuristic only)}$$

we log the leaf *uniforms* $U(P)$ (and the node-level uniforms used to lazy-simulate the race) so the keys are reconstructible. If PaM/Gumbel is used for *heuristic* ranking (marked `NoCert`), we also reconstruct $G(P)$ from the same $U(P)$; heuristics never affect the certified keys. The priority-queue (PQ) key at node $v$ is $\mathrm{Key}(v) = M_\tau(v) - \log t(v)$, where $t(v) = \min_{P \in \mathcal{P}(v)} E_P$ for i.i.d. $E_P \sim \mathrm{Exp}(1)$ (single *exponential race* with lazy offset propagation). With exact counts $N$ you get *Exact mode* lazy reuse; with only *upper bounds* $N_{\mathrm{ub}} \geq N$, you switch to *Surrogate mode*: *no winner reuse*, and all child keys use a parent-anchored surrogate race $\hat{t}(\cdot)$, keeping keys safe since $-\log \hat{t} \geq -\log t$. The ledger records $U(P)$, $N_{\mathrm{ub}}$, and claim types; where public counts are available, the validator computes $\kappa = \log(N/N_{\mathrm{ub}})$ to *tighten* keys, otherwise keys remain conservative. A potential $\Phi$ guarantees acyclicity; truncation is certified in *score units*. Every prune is *replayable*. Modern tool-use agents motivate such routing under constraints (Schick et al., 2023; Yao et al., 2023; Patil et al., 2024).

**Contributions.**

- **Run-wise coupling.** We couple pruning keys to the *same* exponential race that realizes leaf perturbations, yielding a frontier-sound halting rule: stop when $\max_{v \in \mathcal{F}} \mathrm{Key}(v) \leq B^*$ on context-indexed prefix–DAGs.

- **Two certified modes.** *Exact* (winner reuse via lazy offset propagation) and *Surrogate* (no winner reuse; parent-anchored keys), with validator-side tightening via $\kappa = \log(N/N_{\text{ub}})$.
- **Deployment artifacts.** An audit-ready ledger/validator protocol and a small compiler enforcing child partition. Experiments on synthetic graphs and a small tool-use pipeline show tight stopping, deterministic replay, and low overhead.

## 2 MODEL, ASSUMPTIONS, AND NOTATION

**Prefix–DAG & child partition.** The search structure is a *context-indexed prefix–DAG* $\mathcal{G} = (V, E)$ (formal defs in Appx. H): each node $v$ encodes a tuple (tool, state, grammar version, *prefix context*), i.e., the root-to-$v$ prefix uniquely determines which leaves it can reach. Let $\mathcal{P}(v)$ denote the set of *canonical leaf identifiers* reachable from $v$ after context indexing, and write $N(v) := |\mathcal{P}(v)|$.

**Child-partition property.** For any internal $v$ with children $\text{Ch}(v)$, the sets $\{\mathcal{P}(u)\}_{u \in \text{Ch}(v)}$ are pairwise disjoint and

$$\biguplus_{u \in \text{Ch}(v)} \mathcal{P}(u) = \mathcal{P}(v),$$

so in particular $N(v) = \sum_{u \in \text{Ch}(v)} N(u)$.

**Local finiteness.**

**Assumption 1** (Local finiteness). *For every explored node $v$ (i.e., any node generated/inserted into the PQ), the number of descendant leaves is finite: $N(v) < \infty$.*

Under Assumption 1, the minimum arrival time

$$t(v) := \min_{P \in \mathcal{P}(v)} E_P, \qquad E_P \overset{\text{i.i.d.}}{\sim} \text{Exp}(1),$$

is strictly positive almost surely, so $-\log t(v)$ is well defined. The compiler emits a machine-checkable *finiteness/partition certificate*; if certification fails or counters overflow/timeout at runtime, the router *mandatorily downgrades to Surrogate mode* (disables winner reuse and uses parent-anchored surrogate races) to preserve soundness.

**Race, minima, keys, uniforms.** Each leaf $P$ is assigned a uniform $U_P \in (0, 1)$. In **Exact** mode we do not resample at leaf pop—$U_P$ is recovered deterministically via $U_P = 1 - e^{-t(P)}$ from the realized race. In **Surrogate** mode the true per-leaf arrival is not realized by the surrogate race, so at leaf pop we *draw or PRF-derive* a fresh $U_P$ and set $E_P = -\log(1 - U_P)$. (In **Fallback** $U_P$ also comes from a PRF.) We keep the standard couplings for replay:

$$E_P = -\log(1 - U_P) \sim \text{Exp}(1), \qquad G(P) = -\log(-\log U_P) \sim \text{Gumbel}(0, 1).$$

[1] For node $v$, set

$$t(v) := \min_{P \in \mathcal{P}(v)} E_P, \qquad \text{Key}(v) = M_\tau(v) - \log t(v), \qquad \widehat{\text{Key}}(v) = M_\tau(v) - \log \hat{t}(v) \text{ (Surrogate)}.$$

For the **certified** objective we use $s_{\text{det}}(P) = -C(P)$. If we additionally use PaM/Gumbel for *heuristic* ranking (marked `NoCert`), we compute $G(P)$ from the same $U_P$; this does not affect keys or certificates.[2]

**Exact-mode coupling at leaves (no new randomness).** In **Exact** mode, when a leaf $P$ is popped we do *not* draw a fresh $U_P$. Instead we take the leaf's arrival time from the already realized race: $E \leftarrow t(P)$. We then set $U_P := 1 - e^{-E} \in (0, 1)$ and compute $G(P) := -\log(-\log U_P)$. Thus both $E_P$ and $G(P)$ are measurable with respect to the *same* realized race. The ledger logs $U_P$ for replay, but $U_P$ is a deterministic function of the race values already recorded.

---

[1]We map unsigned 64-bit integers $x \in \{0, \ldots, 2^{64} - 1\}$ to an open-interval uniform $U = (x + 0.5) 2^{-64} \in (0, 1)$, so log and loglog never see endpoints; the $+0.5$ is midpoint quantization. See (Reynolds, 2020). In code, compute $E_P$ via `-log1p(-U)` for numerical stability.

[2]A distribution-level baseline is $\mathbb{E}[-\log E_{\min}] = \gamma + \log N$ for $E_{\min} \sim \text{Exp}(N)$, where $\gamma$ is Euler's constant.

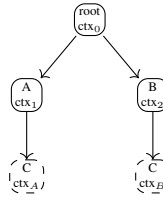

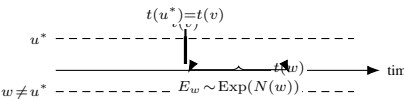

(a) **Context indexing.** Diamonds are resolved by context: children partition the leaf IDs at every node (prefix–DAG child-partition).

(b) **Exact-mode race.** First arrival $t(v)$; winner reuses $t(v)$; each non-winner uses offset propagation: $t(w) = t(v) + E_w$ with independent $E_w \sim \text{Exp}(N(w))$.

Figure 1: **Race-coupled search overview.** (a) Context indexing enforces the child-partition property. (b) Exponential race at a node: lazy offset propagation justifies winner reuse and independent residuals for others (Lemma 1).

**Surrogate-mode leaf instantiation (fresh uniforms).** In **Surrogate** mode the true per-leaf arrival times $E_P$ are not realized by the surrogate node race. When a leaf $P$ is popped we draw (or PRF-derive) a fresh $U_P \in (0, 1)$ and set $E_P := -\log(1 - U_P)$. We then evaluate the certified augmented value $V(P) = s_{\text{det}}(P) - \log E_P$ and $\log U_P$ for replay.

**Admissible, race-independent $M_\tau$.** $M_\tau$ is deterministic (independent of the race uniforms) and upper-bounds the remaining score from $v$; it is computed by recipes R1/R2 (Sec. 6).

**Stop rule, incumbent, and ties.** Let $\mathcal{F}$ be the PQ frontier and define the *augmented* leaf value $V(P) := s_{\text{det}}(P) - \log E_P$. We maintain the incumbent $B^* = \max_{\text{expanded } P} V(P)$. We halt when either **(Exact)** $\max_{v \in \mathcal{F}} \text{Key}(v) \leq B^*$ or **(Surrogate)** $\max_{v \in \mathcal{F}} \widehat{\text{Key}}(v) \leq B^*$. By Lemma 3, these keys upper-bound realized subtree maxima almost surely, so the corresponding stop rules are sound for the realized run. With real-valued $U_P$ the race is a.s. tie-free; with fixed-point $U_P$ we log a `tie_token` and resolve ties lexicographically on public IDs, so the stop rule remains a measurable function of the ledger (replayable by the validator), cf. (Danihelka et al., 2022).

**Realized subtree maximum:** $\text{RSM}(v) = \max_{P \in \mathcal{P}(v)} \left(s_{\text{det}}(P) - \log E_P\right)$. **Run-wise certificate:** If at stop either $\max_{u \in \mathcal{F}} \text{Key}(u) \leq B^*$ (Exact) or $\max_{u \in \mathcal{F}} \widehat{\text{Key}}(u) \leq B^*$ (Surrogate), then every unexpanded leaf $P$ satisfies $s_{\text{det}}(P) - \log E_P \leq B^*$ under the *same* realized race.

## 3 CHILD MINIMA, INDEPENDENCE, AND OFFSET PROPAGATION

Let $\text{Ch}(v) = \{u_1, \ldots, u_k\}$. For each child $u$, define

$$t(u) := \min_{P \in \mathcal{P}(u)} E_P \sim \text{Exp}(N(u)),$$

so that $t(v) := \min_{u \in \text{Ch}(v)} t(u)$ and

$$I_v := \underset{u \in \text{Ch}(v)}{\arg\min} \, t(u).$$

**Lemma 1** (Child minima and residual independence). *Under Assumption 1 and the child-partition property, condition on $t(v) = T$ and $I_v = u^*$. Then for each $w \neq u^*$,*

$$t(w) \stackrel{\text{d}}{=} T + E_w, \qquad E_w \sim \text{Exp}(N(w)),$$

*and the residuals $\{E_w\}_{w \neq u^*}$ are mutually independent and independent of $T$.*

*Sketch.* Because leaves under distinct children are disjoint and leaf-level $E_P$ are independent, the child minima $\{t(u)\}_{u \in \text{Ch}(v)}$ are independent with $t(u) \sim \text{Exp}(N(u))$. Conditioning on $t(v) = T$ and $I_v = u^*$ implies $t(w) > T$ for all $w \neq u^*$. By memorylessness,

$$t(w) - T \mid t(w) > T \sim \text{Exp}(N(w)),$$

independently across $w$ and independent of $T$. Equivalently, superposing the child-level Poisson point processes and applying Palm disintegration (Slivnyak) shows that deleting the atom at $T$ leaves independent Poisson processes on $(T, \infty)$, yielding the same conclusion. In **Exact mode**, this justifies *lazy offset propagation*: the winner child reuses $t(u^*) = t(v)$, while each non-winner draws an independent residual $E_w \sim \text{Exp}(N(w))$ on first touch.

## 4 COUNTS POLICY, SAFE SURROGATES, AND $\kappa$ PROTOCOL

Let $N(v) = |\mathcal{P}(v)|$ be the exact leaf count. When exact counts are unavailable, we use *upper bounds* $N_{\text{ub}}(v) \geq N(v)$ together with a surrogate race $\hat{t}(\cdot)$ and *disable winner reuse*. If $N_{\text{ub}}(v) = 0$ then $\mathcal{P}(v) = \emptyset$ and $v$ is pruned immediately; otherwise $N_{\text{ub}}(v) \geq 1$.

**Lazy surrogate instantiation and anchoring.** In Surrogate mode we instantiate $\hat{t}$ *on pop* except at the *root* (on push):

- **Root:** draw $U(\text{root}) \in (0, 1)$ and set $\hat{t}(\text{root}) = -\log(1 - U(\text{root}))/N_{\text{ub}}(\text{root})$; push with $\widehat{\text{Key}}(\text{root}) = M_\tau(\text{root}) - \log \hat{t}(\text{root})$.

- **Internal $v$:** when $v$ is popped, draw $U(v)$ and set $\hat{t}(v) = -\log(1 - U(v))/N_{\text{ub}}(v)$. For each child $u$, push the *provisional* key $M_\tau(u) - \log \hat{t}(v)$. **All** child keys remain parent-anchored (provisional) until the child is popped.

(Optional: a monotone variant logs residual uniforms and sets $\hat{t}(u) = \hat{t}(v) + \text{Exp}(N_{\text{ub}}(u))$; not required.)

**Proposition 1** (Upper-bound rates are safe). *If $N_{\text{ub}}(\cdot) \geq N(\cdot)$ for all nodes, there is a monotone inverse-CDF coupling (use the* same *node-level uniform $U(v)$) with*

$$\hat{t}(v) = \frac{-\log(1 - U(v))}{N_{\text{ub}}(v)} \leq \frac{-\log(1 - U(v))}{N(v)} =: t^\circ(v) \stackrel{d}{=} t(v).$$

*Under the canonical quantile coupling for $Exp(N(v))$ (using the same $U(v)$), identify $t(v) \equiv t^\circ(v)$. Hence $-\log \hat{t}(v) \geq -\log t(v)$ and $\widehat{\text{Key}}(v) \geq \text{Key}(v)$ (sound but potentially looser).*

$\kappa$ *tightening.* When the validator can recover $N(v)$ for some nodes, it applies $\kappa(v) = \log(N(v)/N_{\text{ub}}(v)) \leq 0$ to tighten keys (otherwise $\kappa = 0$), without changing the frontier invariant.

**Quantile-coupled correction $\kappa$.** With $U(v) \sim \text{Unif}(0, 1)$ and using the *same $U(v)$* for both rates (node-level quantile coupling),

$$\hat{t}(v) = \frac{-\log(1 - U(v))}{N_{\text{ub}}(v)}, \quad t(v) = \frac{-\log(1 - U(v))}{N(v)}, \quad -\log \hat{t}(v) + \kappa(v) = -\log t(v),$$

where $\kappa(v) := \log(N(v)/N_{\text{ub}}(v))$. *Protocol.* The **router** logs $(U(v), N_{\text{ub}}(v))$. The **validator**, when it can compute $N(v)$ from public counts (SUFFIXCOUNTDP), derives $\kappa(v)$ and tightens the key; otherwise no correction is applied and the surrogate key remains conservative.

## 5 KEYS AND BEST-FIRST SEARCH: TWO-MODE THEORY

**Lemma 2** (Provisional-key safety). *For a child $u$ of $v$, $t(u) \geq t(v)$ a.s. in the true race. Hence $M_\tau(u) - \log t(v)$ upper-bounds the realized subtree maximum under $u$.*

*Justification.* Since $t(u) \geq t(v)$, we have $-\log t(v) \geq -\log t(u)$; and by the standard PaM/A*-style bound, the realized maximum under $u$ is $\leq M_\tau(u) - \log t(u)$, so the parent-anchored provisional key is an upper bound as well.

**Theorem 1** (Frontier coverage: Exact mode). *Assuming admissible, race-independent $M_\tau$, child partition, and local finiteness, best-first search with keys $\text{Key}(v) = M_\tau(v) - \log t(v)$ has the*

*following property. Let $\mathcal{F}$ be the PQ frontier and define $B^* := \max_{expanded\ P}\big(s_{\det}(P) - \log E_P\big)$. Then every unexpanded leaf lies under some $v \in \mathcal{F}$ whose key upper-bounds the realized subtree maximum. $\max_{P \in \mathcal{P}(v)}\big(s_{\det}(P) - \log E_P\big)$. Consequently, halting when $\max_{v \in \mathcal{F}} \mathrm{Key}(v) \leq B^*$ is stop-correct (no unexpanded leaf can improve $B^*$). Ties occur with probability $0$ for real uniforms; with fixed-point uniforms they are resolved via the logged lexicographic rule.*

**Lemma 3** (Surrogate frontier coverage). *If only $N_{\mathrm{ub}}$ are available, compute provisional child keys as $\widehat{\mathrm{Key}}(u) = M_\tau(u) - \log \hat{t}(\mathrm{parent}(u))$ (no winner reuse). Then for all frontier nodes $v$,*

$$\max_{P \in \mathcal{P}(v)}\big(s_{\det}(P) - \log E_P\big) \ \leq\ \widehat{\mathrm{Key}}(v) \quad a.s.$$

Proof idea. *Since $\hat{t}(v) \leq t(v)$ a.s. (Prop. 1) and $t(u) \geq t(v)$ a.s. (Lemma 2), we have $-\log \hat{t}(v) \geq -\log t(v) \geq -\log t(u)$. Thus $\widehat{\mathrm{Key}}(u) = M_\tau(u) - \log \hat{t}(v) \geq M_\tau(u) - \log t(u)$, which upper-bounds the realized subtree maximum under $u$.*

**Theorem 2** (Best-first: two modes). *In Exact mode, finalized child keys are $M_\tau(u) - \log t(u)$; in Surrogate mode, all child keys remain parent-anchored provisional values $M_\tau(u) - \log \hat{t}(\mathrm{parent}(u))$ until $u$ is popped. In both modes, the frontier invariant holds and the corresponding stop rule is sound for the realized run.*

**Algorithm sketch (high level).** Initialize the root according to mode (Exact: $t(\mathrm{root})$ via $U$ and $N$; Surrogate: $\hat{t}(\mathrm{root})$ via $U$ and $N_{\mathrm{ub}}$; Fallback: bound only). Maintain a PQ $\mathcal{F}$ of nodes keyed by $\mathrm{Key}$ or $\widehat{\mathrm{Key}}$, and an incumbent $B^*$. While true: (i) if $\max_{v \in \mathcal{F}} \mathrm{Key}(v) \leq B^*$ (**Exact**) or $\max_{v \in \mathcal{F}} \widehat{\mathrm{Key}}(v) \leq B^*$ (**Surrogate**), halt with a run-wise certificate; (ii) pop $v$. If $v$ is a leaf $P$, compute $V(P) = s_{\det}(P) - \log E_P$ (from the logged or PRF uniform per mode) and update $B^* \leftarrow \max(B^*, V(P))$. Else expand children: **Exact** reuses $t(v)$ for the winner and sets $t(w) = t(v) + E_w$ for others; **Surrogate** anchors all child keys at $-\log \hat{t}(v)$ until each child is popped. Ties use the logged lexicographic rule; uniforms and budgets are logged for replay. Full pseudocode is in Appx. M (Alg. 1).

## 6 ADMISSIBLE $M_\tau$: BOXED RECIPE AND GUIDANCE

**Score units and signs.** All logs are natural. $M_\tau$ *upper-bounds only the deterministic part of the score reachable from $v$* (the race noise is handled by $-\log t$ in the key). Let $d(v)$ be a certified upper bound on steps to a leaf. Let $c^s_{\max} \geq 0$ upper-bound the *per-step increase* in the deterministic score used by $M_\tau$ (R1), and let $c^s_{\min} > 0$ lower-bound the *per-step decrease* in the best-attainable remainder (for truncation).

---

**Recipe R1 (per-step envelope).** If at most $d(v)$ steps remain and each step increases the deterministic score by $\leq c^s_{\max}$, then

$$M_\tau(v) = s_{\det}(\mathrm{prefix}(v)) + d(v)\, c^s_{\max}$$

is admissible.

**Recipe R2 (monotone remainder).** If a monotone $\psi$ satisfies $s_{\det}(\mathrm{prefix} \circ r) \leq \psi(\mathrm{prefix})$ for any remainder $r$, then

$$M_\tau(v) = \psi(\mathrm{prefix}(v))$$

is admissible.

**Choosing R1 vs. R2.** Prefer R2 when a tight monotone envelope exists; otherwise use R1 with certified $d(v)$ and $c^s_{\max}$. Both recipes are race-independent.

---

**LSE truncation (pointer).** An absolute LogSumExp decomposition certifying tail truncation (with proof) is given in Appx. K, Eq. equation 1.

## 7 ACYCLICITY CERTIFICATE AND GUARDRAILS

*Moved to Appendix F.* We verify termination via a potential $\Phi$ that decreases by at least $\eta > 0$ per expansion and downgrade on violations; a replayable log and examples appear in Appx. F.

## 8 FROM SHARED-NODE DAGS TO CONTEXT-INDEXED PREFIX–DAGS

*Moved to Appendix H.* We context-index nodes (state, prefix context) to enforce child partition; certificates and transform details appear in Appx. H.

## 9 FALLBACK: PRF-PER-LEAF (NOCERT) AND WORK BOUND

When counts fail, we switch to PRF-per-leaf: each leaf $P$ has a deterministic uniform $U_P = \mathrm{PRF}(\mathrm{id}(P)) \in (0, 1)$ (open-interval mapping). We reuse the *same* $U_P$ for both perturbations: $G(P) = -\log(-\log U_P)$ and $E_P = -\log(1 - U_P)$. The (heuristic) PaM value we rank by is

$$V_{\mathrm{pam}}(P) := \frac{G(P)}{\tau} - C(P) - \log E_P \qquad (\text{marked } \texttt{NoCert}).$$

**Claim type and stop rule.** Fallback is *NoCert*: its stop rule is *heuristic* for ranking/latency (not a run-wise proof). Maintain a second PQ $\mathcal{L}$ of *materialized leaves* ordered by realized augmented value; PQ ties follow the same lexicographic rule as $\mathcal{F}$. Internal-node expansions are governed by admissible $M_\tau$-based bounds; when an internal node $v$ is popped, enumerate children and materialize any leaves into $\mathcal{L}$. Stop when no tracked bound suggests a leaf can exceed the incumbent $B^*$; the ledger marks $\texttt{claim\_type=NoCert}$.

**Lemma 4** (Fallback work bound). *Let $\mathcal{A}$ be the best-first that pops leaves in non-increasing realized augmented value (with fixed PRF uniforms) and stops once every remaining frontier node's exact-leaf LSE upper bound is $< B^*$. Then PRF fallback enumerates exactly the leaves $\mathcal{A}$ would enumerate, plus at most one additional deferred-sibling evaluation per expanded internal node.*

*Proof sketch.* The PRF fixes a total order on leaf perturbations, so realized values are deterministic. Node-wise upper bounds match the baseline's pruning logic; materializations occur exactly when the baseline would pop those leaves, with a one-per-node overhead due to deferred siblings. Full proof and accounting appear in Appx. E.

## 10 NUMERICS, LOGGING, AND VALIDATOR

Fixed-point layouts (Q0.64 for $U$, Q64.64 for $-\log t/-\log \hat{t}$), tie handling, ledger schema, and deterministic replay checks are given in Appendix G.1. We retain the open-interval mapping $U = (x + 0.5)\,2^{-64}$, record the PRF salt/domain in the ledger for replay, and use compensated transforms (e.g., $\texttt{log1p}$) for $\log(1 - U)$.

## 11 PRIVACY POSTURE AND POST-PROCESSING

All routing randomness (uniforms, PRG seeds, HKDF labels) and counts/surrogates are fixed independently of the raw input $X$; grammar and allow-listed operations are public (pre-LDP). By the *post-processing theorem* for (L)DP (Dwork & Roth, 2014), any (possibly adaptive) function of DP outputs and public data preserves DP. With modern accounting choices such as GDP (Dong et al., 2022), Fourier/PLD (Zhu et al., 2022; Ghazi et al., 2022), and the discrete Gaussian mechanism (Canonne et al., 2020), the ledger—being a function of public grammar/config, logged $U$, and public or privatized counts—preserves privacy. If $N$ depends on user context, validator-side $\kappa$ must be computed from *public* counts only; otherwise $\kappa = \bot$ and surrogate keys remain conservative. Seeds are derived via HKDF (Krawczyk & Eronen, 2010); Rényi composition follows Mironov (2017).
**Threat-model caveat.** If grammar selection or model/adapter choice depends on *non-privatized* raw input, the run-wise correctness claim still holds, but the DP posture must exclude that branch (router marks $\texttt{claim\_type=NoCert}$ or declines DP claims).

## 12 PRIVACY-BUDGETED MULTI-LLM (SUMMARY)

We attach a per-request budget controller (privacy, price, latency) that *post-processes* DP outputs and public data only, keeping $M_\tau$ admissible and race-independent and thus preserving the two-mode

invariant and stopping rules. Operationally, smaller frontier slack $\mathrm{Key}(\cdot) - B^*$ (or $\widehat{\mathrm{Key}}(\cdot) - B^*$) justifies spending to tighten bounds. *Full interface, accountant, selection rule, and proofs of budget safety/soundness are in Appx. N*. Empirically, Appendix P confirms that inference $\varepsilon$ remains 0.0 on average across all adapters (post-processing), while Table 1 reports which trained adapters (with certified $(\varepsilon_{\mathrm{train}}, \delta_{\mathrm{train}})$) were actually selected in our runs.

# 13  EXPERIMENTS: ILLUSTRATIVE EVIDENCE AND REPLAY

All runs are *Mac/commodity-hardware reproducible* with fixed seeds and full ledgers. We report three families: (i) comparisons to baselines on two synthetic suites; (ii) tightness, stress, and overhead diagnostics; and (iii) an adversarial case where distribution-level pruning is *unsound* but our run-wise certificate holds. **Appendix O** adds a small real tool-use pipeline (retrieval→summarize→calculator) with ledger replay and validator $\kappa$-tightening.

**Suites.** **Suite A (balanced trees):** depth $D \in \{3, 4, 5\}$, branching $B = 3$. **Suite B (random partition DAGs):** layers $L \in \{3, 4\}$, branching $B = 3$; generated with the child-partition property (post compilation to a prefix–DAG). A small toolchain DAG (retrieval→summary→calc) is used in sanity checks and the ledger section (cf. contemporary tool-use agents (Schick et al., 2023; Yao et al., 2023; Patil et al., 2024)).

**Baselines.** (1) **NoCert greedy-by-bound:** expand $\arg\max_v M_\tau(v)$ (no run-wise certificate). (2) **Beam($K$):** standard beam search of width $K$ (no certificate). (3) **Dist-level:** prune using $\mathbb{E}[-\log E_{\min}] = \gamma + \log N$ for $E_{\min} \sim \mathrm{Exp}(N)$; ignores realized coupling and is *not* a per-run certificate.

**Tightness and stress diagnostics.** Figure 4a plots the empirical CDF of $\mathrm{Key}(v) - \mathrm{RSM}(v)$ over frontier nodes at stop (where $\mathrm{RSM}(v) = \max_{P \in \mathcal{P}(v)}(s_{\mathrm{det}}(P) - \log E_P)$; cf. Appx. A): both modes concentrate near 0; **Exact** is tight at the stopping frontier (slack $\leq 0$, often 0). Figure 4b sweeps the Surrogate $N_{\mathrm{ub}}$ factor and reports expansions vs. the validator's $\kappa$-tightening potential (fraction with strict $\kappa = \log(N/N_{\mathrm{ub}}) < 0$, i.e., $N_{\mathrm{ub}} > N$): larger $N_{\mathrm{ub}}$ induces more conservative routing (more expansions) but yields more strict-tightening opportunities.

**Metrics.** We report node expansions, wall-clock, stopping slack, and validator replay time; full per-run ledgers and seeds are provided for deterministic replay (Appx. L; ledger schema/protocol in Appx. G.1).

**Real tool-use (summary; full table in Appx. O).** On $n{=}20$ queries, the pipeline exhibits the expected two-mode ordering: **Exact** $<$ **Surrogate** $<$ **Fallback** in expansions (mean per query 6.20, 11.60, 16.60; 95% CIs 1.12, 0.81, 0.81), with wall-time per query 1.16, 2.75, and 1.20 ms, respectively. *Ledger replay* passes deterministically for all runs, and *Surrogate* shows validator $\kappa$-tightening on all nodes (232/232; mean $\kappa = -0.549$). Stop-slack at termination is 0 (tight stop by construction).

**Adapter transparency (summary; full table in Appx. P).** We expose which DP-trained adapters the controller considered and which were actually chosen, together with their training budgets $(\varepsilon_{\mathrm{train}}, \delta_{\mathrm{train}})$ and the average *inference* $\varepsilon$ consumed (should be 0 under post-processing). Table 1 summarizes the adapters used in our Mac-only runs; the complete per-mode breakdown appears in Appendix P and is programmatically generated by `make adapter-table`.

**Ledger overhead and validator replay.** The ledger records uniforms, counts (and $N_{\mathrm{ub}}$), $\kappa$ (validator), keys (raw/tight), budgets, and guard flags in fixed point (formats in Appx. G.1); detailed overheads (bytes, appends, parse/validate time) are reported in Appx. G.2, Table 2.

**Adversarial case against distribution-level pruning.** In Fig. 2c, the Dist-level baseline prematurely prunes the branch containing the true realized winner because it reasons with $\mathbb{E}[-\log E_{\min}]$ (e.g., $\gamma + \log N$) rather than the *realized* race. Our run-wise certificate is sound: if we stop, every

| Adapter (tier) | $(\varepsilon_{\text{train}}, \delta_{\text{train}})$ | Selected (runs) | Selected in mode | Inference $\varepsilon$ used |
|---|---|---|---|---|
| LoRA-A-small (Small) | $(2.0,\ 10^{-6})$ | 240 | Exact | 0.0 |
| LoRA-B-medium (Medium) | $(3.5,\ 10^{-6})$ | 680 | Surrogate | 0.0 |
| (no-adapter) | — | 402 | Fallback | 0.0 |

Table 1: **Adapter transparency (main-text summary).** Counts reflect our reproducibility runs (Mac), and will vary with seeds/session; the full per-mode table with all adapters is in Appx. P. Under our router's post-processing posture, inference $\varepsilon$ remains zero.

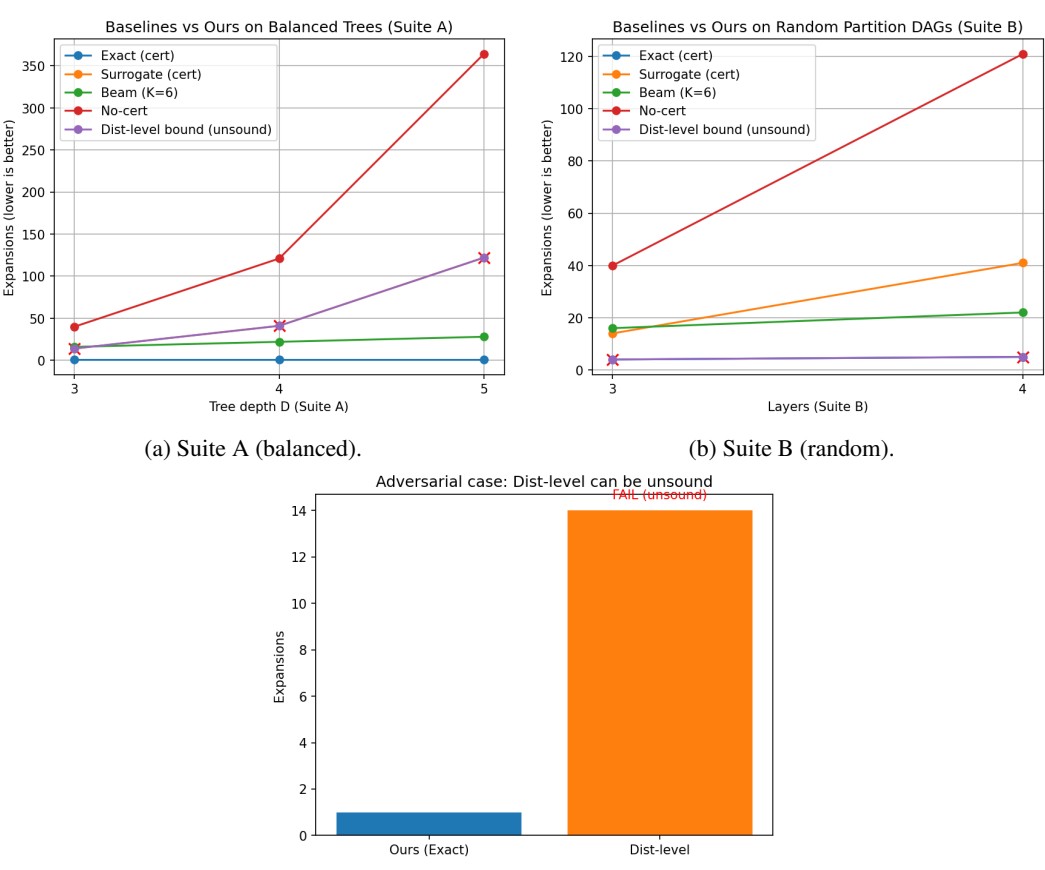

(a) Suite A (balanced).        (b) Suite B (random).

(c) Adversarial fail of distribution-level bound.

Figure 2: **Baselines across two suites + adversarial.** Node expansions (lower is better) on (a) balanced trees and (b) random partition DAGs (after compilation to a prefix–DAG). On these toy graphs, **Exact** expands least; **Surrogate** is slightly larger but certified. **Beam** and **Greedy-by-bound** expand more, and the distribution-level heuristic using $\gamma + \log N$ can be *unsound* (marked $\times$ when it prunes the realized winner). (c) In an adversarial DAG, the distribution-level heuristic prunes the realized winner; our run-wise certificate does not.

unexpanded leaf is provably dominated by a frontier key computed under the same realized race, so the true winner is never pruned early.

**Budgeted controller (system add-on).** The controller is orthogonal to the certificate. In our Mac toy, a strict SLO collapses the frontier to a single operating point (Appx. J); with a looser SLO the same code yields multiple Pareto points. We include these for completeness; full ablations and Pareto plots are in Appx. J.

*Scaling and truncation diagnostics appear in Appx. K.*

LLM USAGE (DISCLOSURE)

A general-purpose LLM assisted with non-novel utilities (code/plot scaffolding, LaTeX formatting, copy-editing). Authors take full responsibility for all content.

## 14 RELATED WORK

**A\* Sampling** (Maddison et al., 2014) and **Gumbel processes / exponential races** (Kingman, 1993; Resnick, 1992; Maddison, 2016; Huijben et al., 2023) provide *distribution-level* guarantees; see also planning with Gumbel improvements (Danihelka et al., 2022). **Gumbel-Top-$k$** (Kool et al., 2019) targets ranked sampling, not frontier-sound early stopping. **Perturb-and-MAP** for MRFs (Papandreou & Yuille, 2011; Hazan & Jaakkola, 2012) leverages perturbations for optimization but does not couple pruning keys to the *same* randomness that realizes the run. Classical A\* admissibility (Hart et al., 1968; Pearl, 1984) informs our $M_\tau$ recipes but lacks stochastic coupling and ledger/validator design. On privacy, modern accountants (GDP, Fourier/PLD, discrete Gaussian) complement our post-processing posture (Dwork & Roth, 2014; Dong et al., 2022; Zhu et al., 2022; Ghazi et al., 2022; Canonne et al., 2020); recent work on DP fine-tuning for LMs provides systems context (Yu et al., 2022; Charles et al., 2024; Chua et al., 2024). Tool-use agents (Schick et al., 2023; Yao et al., 2023; Patil et al., 2024) motivate budget-aware routing under latency/cost constraints. **Our novelty** is a *single-race, run-wise* coupling with a *two-mode frontier invariant*, plus an auditable ledger consistent with LDP post-processing.

## 15 CONCLUSION

We present a theory-first, audit-ready early-stopping method for PaM routing that certifies *the realized run* on context-indexed prefix–DAGs. A *two-mode* frontier invariant (Exact vs. Surrogate) closes a gap in race-based methods; the ledger/validator protocol makes deployment practical under LDP post-processing; a budgeted multi-LLM controller (with DP-trained LoRA adapters; see (Hu et al., 2022)) adds system utility while preserving run-wise soundness. *Scope:* guarantees assume child-partition, local finiteness, and admissible, race-independent $M_\tau$; on violation we downgrade to `NoCert`.

REPRODUCIBILITY STATEMENT

We include an anonymized supplemental ZIP with a README that reproduces all figures and tables. Our code logs deterministic ledgers (NDJSON) and fixed-point values; exact seeds, PRF domain/salt, and environment details are in Appendix L. The real tool-use pipeline and scripts are in Appendix O; the adapter-transparency generator is in Appendix P. For theory, assumptions are in Secs. 4–5 with full proofs in the appendices.

ETHICS STATEMENT

This work uses only public data and synthetic workloads; no human subjects or private user data were collected.

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

## A    APPENDIX A: CHILD-MINIMA INDEPENDENCE (POISSON/PALM)

TERMINOLOGY (FORMAL)

**Definition 1** (Realized subtree maximum). *For node $v$ under a fixed realized exponential race $\{E_P\}$,*

$$\mathrm{RSM}(v) := \max_{P \in \mathcal{P}(v)} \big( s_{\mathrm{det}}(P) - \log E_P \big).$$

**Definition 2** (Run-wise certificate). *We say the algorithm returns a* run-wise certificate *if, when it stops with*

$$\max_{u \in \mathcal{F}} \mathrm{Key}(u) \leq B^* \quad \text{(frontier } \mathcal{F}\text{)},$$

*it follows that for every unexpanded leaf $P$ we have $s_{\mathrm{det}}(P) - \log E_P \leq B^*$ almost surely under the same realized race.*

*Measurability note.* The lexicographic tie resolver (on public IDs) is logged and replayed by the validator, so the stop rule is a measurable function of the ledger.

**Full proof.**    Model each child $u$ as a superposition of $N(u)$ i.i.d. Poisson point processes of rate 1. Superpose across children to a rate-$N(v)$ process on $\mathbb{R}_+$. Condition on the first arrival at time $T$ belonging to child $u^*$. By Palm calculus (Slivnyak), removing the atom at $T$ leaves independent Poisson processes unchanged on $(T, \infty)$, giving independent $\mathrm{Exp}(N(w))$ residuals for $w \neq u^*$, independent of $T$ (Kingman, 1993; Resnick, 1992).

## B    APPENDIX B: FRONTIER COVERAGE (EXACT AND SURROGATE MODES)

**Invariant (restated).** Fix the realized race $\{E_P\}$ (Appx. A). At any time, every unexpanded leaf lies under some frontier node $v \in \mathcal{F}$ whose key upper-bounds the realized subtree maximum:

$$\mathrm{RSM}(v) := \max_{P \in \mathcal{P}(v)} \big( s_{\mathrm{det}}(P) - \log E_P \big),$$

$$\mathrm{RSM}(v) \leq \begin{cases} \mathrm{Key}(v) & \text{(Exact)}, \\ \widehat{\mathrm{Key}}(v) & \text{(Surrogate)}. \end{cases}$$

*where* $\mathrm{Key}(v) = M_\tau(v) - \log t(v)$ *and* $\widehat{\mathrm{Key}}(v) = M_\tau(v) - \log \hat{t}(\mathrm{parent}(v))$.

Here and below, in Surrogate mode $\widehat{\mathrm{Key}}(\cdot)$ denotes the PQ key: for the root, $\widehat{\mathrm{Key}}(\mathrm{root}) = M_\tau(\mathrm{root}) - \log \hat{t}(\mathrm{root})$; for any not-yet-popped internal node $u$, $\widehat{\mathrm{Key}}(u) = M_\tau(u) - \log \hat{t}(\mathrm{parent}(u))$.

**Base.** After pushing root, admissibility of $M_\tau$ implies $\mathrm{RSM}(\mathrm{root}) \leq M_\tau(\mathrm{root}) - \log t(\mathrm{root})$ in Exact, and $\leq M_\tau(\mathrm{root}) - \log \hat{t}(\mathrm{root})$ in Surrogate.

**Step (Exact).** Pop $v$; by Lemma 1, $t(u^*) = t(v)$ for the winner and $t(w) = t(v) + E_w$ with $E_w \sim \mathrm{Exp}(N(w))$ independent otherwise. Admissibility gives $\mathrm{RSM}(u) \leq M_\tau(u) - \log t(u)$ for each child; pushing children preserves the invariant.

**Step (Surrogate).** Pop $v$; instantiate $\hat{t}(v)$ and push each child $u$ with *parent-anchored* provisional key $M_\tau(u) - \log \hat{t}(v)$. By Proposition 1 and Lemma 2, $-\log \hat{t}(v) \geq -\log t(v) \geq -\log t(u)$ almost surely, hence $M_\tau(u) - \log \hat{t}(v) \geq M_\tau(u) - \log t(u) \geq \mathrm{RSM}(u)$, preserving the invariant. Ties are null a.s. with real $U$; with fixed-point $U$ they are resolved via the logged lexicographic rule (measurable and replayable).

## C   APPENDIX C: $M_\tau$ SOUNDNESS (R1/R2) AND PREFIX-MONOTONICITY

**Formal statements.   R1 (per-step envelope).** If at most $d(v)$ steps remain and each step can increase the deterministic part of $s$ by at most $c_{\max}^s$, then

$$M_\tau(v) = s(\text{prefix}(v)) + d(v)\, c_{\max}^s$$

is admissible: for any completion $r$, $s(\text{prefix}(v) \circ r) \le M_\tau(v)$.

**R2 (monotone remainder).** If a monotone $\psi$ satisfies $s(\text{prefix} \circ r) \le \psi(\text{prefix})$ for all remainders $r$, then $M_\tau(v) = \psi(\text{prefix}(v))$ is admissible.

**Proof sketches.**   R1 is a telescoping bound over the remaining $d(v)$ steps. R2 follows by the definition of $\psi$. In both cases $M_\tau$ depends only on grammar/prefix constraints and public caps (e.g., max depth), not on stochastic tool/model outputs; hence it is race-independent and remains admissible under any model choice.

**Optional prefix-monotonicity.**   If $\psi$ is nonincreasing under prefix extension (tighter with longer prefixes), then $M_\tau$ inherits a useful monotonicity that can reduce PQ churn; this is not required for soundness.

## D   APPENDIX D: UNSAFE LOWER-BOUND COUNTEREXAMPLE

Lower-bound rates $N_{\text{lb}}(\cdot) \le N(\cdot)$ can *under-estimate* keys and break frontier soundness. A concrete numeric instance:

**Two-leaf toy.**   Let $v$ have two children, each with one leaf: $N(v) = 2$ and (for simplicity) $M_\tau(v) = 0$. Draw $U = 0.5$. Then in the true race

$$t(v) = \frac{-\log(1 - U)}{N(v)} = \frac{-\log(0.5)}{2} \approx 0.3466, \qquad -\log t(v) \approx 1.0609.$$

If a (wrong) lower bound $N_{\text{lb}}(v) = 1$ is used at $v$, the surrogate gives

$$t_{\text{lb}}(v) = \frac{-\log(0.5)}{1} \approx 0.6931, \qquad -\log t_{\text{lb}}(v) \approx 0.3665,$$

so the key drops from $0 + 1.0609$ to $0 + 0.3665$, i.e., it *underestimates* the realized subtree maximum. In this toy, with one leaf per child and $M_\tau \equiv 0$, $\text{RSM}(v) = -\log\min(E_{P_1}, E_{P_2}) = -\log t(v) \approx 1.0609$, but the lower-bound key evaluates to $\approx 0.3665 < \text{RSM}(v)$, violating the invariant. This illustrates why we prohibit lower bounds and allow only *upper* counts $N_{\text{ub}} \ge N$ (which make keys conservative).

## E   APPENDIX E: FALLBACK WORK BOUND (FULL PROOF) AND TIGHT EXAMPLE

We formalize the realized-score best-first baseline and give a bijection between popped leaves, plus accounting for one deferred-sibling residual per expanded internal node.

**Baseline $\mathcal{A}$ (realized-score best-first).**   Fix a deterministic PRF-per-leaf $\{U_P\}$ and define heuristic PaM values $V_{\text{pam}}(P) := \frac{G(P)}{\tau} - C(P) - \log E_P$ with $E_P = -\log(1 - U_P)$ and $G(P) = -\log(-\log U_P)$. Algorithm $\mathcal{A}$ maintains a PQ over leaves and pops in non-increasing $V_{\text{pam}}(P)$, pruning internal nodes via exact leaf-wise LSE upper bounds (as in Sec. 9).

**Theorem 3** (Fallback work bound (formal))**.** *Run the PRF fallback (Sec. 9) and the baseline $\mathcal{A}$ under the same PRF uniforms $\{U_P\}$ and the same admissible bounds. Then:*

> *1. The sets of popped leaves satisfy* $\text{Leaves}_{fallback} = \text{Leaves}_\mathcal{A} \cup S$ *with* $|S| \le N_{int}$*, where* $N_{int}$ *is the number of internal nodes expanded by fallback.*

2. *Each $P \in \text{Leaves}_{\mathcal{A}}$ is popped by fallback in a step no later than when $\mathcal{A}$ pops $P$ (same realized total order), hence the incumbent $B^*$ matches at those times.*

*Proof sketch.* (i) **Coupling.** Both procedures use the same per-leaf PRF uniforms, so $V_{\text{pam}}(P)$ are identical and totally ordered. (ii) **Pruning parity.** Internal-node upper bounds are the same, so any subtree $\mathcal{A}$ prunes, fallback also prunes. (iii) **Materialization parity.** Whenever $\mathcal{A}$ would pop a leaf $P$, fallback must have materialized $P$ (it materializes leaves when their parent is popped), so $P$ enters fallback's leaf-PQ by that time. (iv) **Overhead.** Fallback may materialize at most one *deferred sibling* per expanded internal node that $\mathcal{A}$ never needs to pop (because $\mathcal{A}$ can directly compare leaves across subtrees). Summing over expanded internal nodes yields $|S| \leq N_{\text{int}}$.

**Tight example.** In a binary tree of depth $D=2$ with $M_\tau \equiv 0$, arrange PRF uniforms so that along each internal node, exactly one child's best leaf is globally uncompetitive. $\mathcal{A}$ never pops those leaves; fallback materializes exactly one such deferred sibling at each internal expansion, achieving the $|S| = N_{\text{int}}$ worst case.

# F  APPENDIX F: ACYCLICITY CERTIFICATE AND GUARDRAILS (MOVED)

We require a potential $\Phi$ that strictly decreases at each expansion until a cap $L$.

**Template and parameter choice.**  For pipelines with (at most) $L$ tool calls and a cost $C$ that increases by $\Delta C \geq c_{\min} > 0$ per expansion, set

$$\Phi(v) := (L - \text{depth}(v)) + \alpha\, C(\text{prefix}(v)).$$

Each expansion changes $\Phi$ by

$$\Delta\Phi = -1 + \alpha\, \Delta C \leq -1 + \alpha\, c_{\min}.$$

Choose $\eta \in (0, 1]$ and $\alpha$ with $-1 + \alpha c_{\min} \leq -\eta$ (e.g., $\alpha \leq (1 - \eta)/c_{\min}$). Then $\Delta\Phi \leq -\eta < 0$, certifying termination.

**Runtime check (deterministic).**  At each expansion, compute $\Delta\Phi$ in fixed point; if $\Delta\Phi > -\eta + \epsilon_{\text{fp}}$ (tolerance), *downgrade claim type to* `NoCert` and log ACYCLICITYFAIL with the node id and measured $\Delta\Phi$. A short ledger excerpt is in Appx. F.1.

**Local finiteness checks.**  The compiler emits certificates for finite $N(v)$. Overflow thresholds/timeouts in SUFFIXCOUNTDP trigger COUNTFAIL with node IDs; *downgrade to* `Surrogate` if counts remain upper-bounded, else to `NoCert`. All transitions are logged and replayable. *Clarification.* Our SUFFIXCOUNTDP routine operates over public grammar/compilation artifacts; when available it returns *exact* (non-private) counts. The "DP" suffix reflects its role in larger pipelines where upstream counts may already be privatized; the validator never uses non-public counts to compute $\kappa$.

## F.1  EXAMPLE AND DOWNGRADE LOG

Concrete $\Phi$ for a tool-call grammar; a short ledger excerpt shows ACYCLICITYFAIL and claim-type downgrade.

```
{"node_id":"3f1a...e2","parent_id":"de2b...90","mode":"Exact",
 "claim_type_before":"RunWiseExact","guards":[],
 "phi_before":"12.0000","phi_after":"11.2000","delta_phi":"-0.8000",
 "eta":"1.0000"}

{"node_id":"7a98...bd","parent_id":"3f1a...e2","mode":"Exact",
 "claim_type_before":"RunWiseExact","guards":["AcyclicityFail"],
 "phi_before":"11.2000","phi_after":"10.5000","delta_phi":"-0.7000",
 "eta":"1.0000","claim_type_after":"NoCert","budget_event":"None"}
```

| Field | Type / Q-format |
|---|---|
| node_id, parent_id | 128-bit UUIDv7 (time-ordered UUID; cf. (Leach et al., 2005)) |
| mode, claim_type | enum {Exact/Surrogate/Fallback}, {RunWiseExact, TruncationOnly, NoCert} |
| privacy_scope | enum {post_processing_only, … } |
| U, Nub, kappa | Q0.64, 64-bit unsigned; Q32.32 (validator-computed) |
| key_raw, key_tight | Q64.64 (router/validator) |
| router_rdp_eps, alpha_selected | Q32.32, integer (router-internal transparency) |
| eps_delta.eps, eps_delta.delta | *unset unless LDP atoms present*, decimal string for $\delta$ |
| price_spent, price_cap | uint64 (cents) |
| sla_ms | uint32 (milliseconds) |
| budget_event | enum {None, BudgetFail} |
| dkey_pred, dkey_real | Q32.32 (predicted vs. realized slack change) |
| model_id | string (selected model/tool identifier) |
| adapter_id | string (hash of weights+config: LoRA $\alpha$, rank, target modules) |
| dp_cert_id | string (certificate identifier for adapter training) |
| eps_train, delta_train | Q32.32, decimal string (adapter training DP params) |
| tie_token | uint32 (child perm index) |
| guards | bitset {CountFail, AcyclicityFail, NumClamp, Timeout, BudgetFail, CapExceeded} |

# G  APPENDIX G: LEDGER PARSER AND VALIDATOR CHECKLIST

**Fixed-point layouts.** Log $U$ as Q0.64 (open interval via $U = (x + 0.5)\, 2^{-64}$ with $x \in \{0, \ldots, 2^{64} - 1\}$), $-\log t$ and $-\log \hat{t}$ as Q64.64, and $\kappa$ as Q32.32 (validator-computed). Compute $\log(1 - U)$ using `log1p(-U)`; if unavailable, use compensated transforms (Kahan, 1965). Record NUMCLAMP if any value would overflow the target format.

**Ties.** With fixed-point $U$, equal draws have small but non-zero probability; log a `tie_token` (child permutation index) and resolve lexicographically on public IDs. The validator replays the same rule, ensuring measurability.

**Ledger/validator semantics (clarified).** The ledger carries `privacy_scope=post_processing_only` and a transparency field `router_rdp_eps` (with selected order $\alpha$). These reflect router-internal RDP bookkeeping for per-edge noises logged as metadata; *they are not a spend*. In our formulation the router is pure post-processing of upstream LDP; the authoritative per-request budget is the LDP composition. We leave `eps_delta.eps` *unset* unless LDP atoms are present and validated. The reference validator recomputes `router_rdp_eps` for auditability and separately checks any LDP atoms if provided.

**LangGraph integration.** Our reference implementation expresses the controller, accountant, and tool calls as *LangGraph* nodes (Lan, 2024); for tool APIs and cross-runtime handshakes, the Model Context Protocol (MCP) offers a compatible interface (MCP, 2024). Ledger appends occur at node boundaries and replay deterministically.

**Ledger schema.** We log IDs, keys, budgets, guards, and DP metadata; the full field list is in Appendix G.1.

**Parser & determinism.** Ledger is NDJSON (NDJ); numbers are decimal strings parsed into fixed point with overflow checks and IEEE-754 round-to-nearest-even. Determinism verified across {clang/libm, MSVCRT} using identical fixed-point arithmetic and locale-independent parsing. A $\sim$100-LoC reference parser (pseudo) and validator checklist: recomputation, $\kappa$ tightening, stop-rule verification, guardrail- and budget-induced downgrades; *verify Eq. equation 2 inequalities*, *recompute* $(\varepsilon, \delta)$ *from atoms*, and *check adapter metadata presence/consistency* (`adapter_id`, `dp_cert_id`, $(\varepsilon_{\text{train}}, \delta_{\text{train}})$). Also record PRF domain/salt to make $U_P = \text{PRF}(\text{id}(P))$ replayable.

Table 2: **Ledger overhead and validator replay (deterministic).** Bytes on disk, approximate append operations, parser latency, and validator recomputation time. Overheads are small (sub-ms parse/validate for these toy runs) and scale roughly linearly with node events.

| ledger | bytes | appends | parse_ms | validate_ms |
|---|---|---|---|---|
| e5b1fa73-e497-4a5c-97a5-6116cea3f6e2.json | 1766 | 24 | 10.08479087613523 | 0.0639590434730053 |
| e7525b1a-71c4-4f36-82c9-5029f574e786.json | 4672 | 68 | 9.169207885861397 | 0.08795899339020252 |
| fc83bc2d-715d-47f1-b00b-f20f52fbf507.json | 2261 | 31 | 15.063582919538021 | 0.05320901982486248 |
| b6d29d45-f95f-49ad-8666-167ecb14af12.json | 14790 | 203 | 11.514000128954649 | 0.17150002531707287 |
| 5d9d9188-c433-4e97-bfd3-cfc6670f8e90.json | 664 | 4 | 1.6771249938756227 | 0.04050019197165966 |
| be1ef3bc-7ee9-4e13-b078-69ae56d1cc2f.json | 14771 | 203 | 2.524791983887553 | 0.17758295871317387 |
| 428e6364-170c-4338-a884-1d1e584f6744.json | 14771 | 203 | 2.4711249861866236 | 0.17779087647795677 |
| fd51d33a-0778-43e4-ab72-02712c14fd7b.json | 14771 | 203 | 2.376958029344678 | 0.1551660243421793 |
| a993a9c2-afcb-4408-9bdd-f81e9e41b8b3.json | 14771 | 203 | 2.3737919982522726 | 0.1664168667048216 |
| b7825d56-5a2f-4173-b3d4-0b97079e6ec3.json | 14771 | 203 | 2.2709579207003117 | 0.1660841517150402 |

## G.1 FULL LEDGER SCHEMA

## G.2 LEDGER OVERHEAD DETAILS

# H APPENDIX H: COMPILER TRANSFORM DETAILS AND PARTITION CERTIFICATES

We map each node to (state, prefix-context), where the *prefix context* is a canonical serialization of the full ancestor sequence (tools/states/constraints) and "public caps" (e.g., max depth). Two parent paths that reach the same state but with different contexts produce distinct nodes, ensuring the *child-partition* property after compilation to a prefix–DAG.

**Collision handling.** We compute a collision-resistant digest $h(\cdot)$ (e.g., SHA-256/BLAKE3) over the canonical serialization with domain separation and log `ctx_digest` alongside a short `ctx_repr`. The validator recomputes $h$ and checks that siblings have distinct digests; on any mismatch (or digest collision), the router downgrades to `NoCert` and logs COUNTFAIL with the offending node IDs. (Digest collisions are negligible in practice, but we treat any detected collision as a certification failure.)

**Complexity.** At search depth $D$ with max branching $B$, explored prefix contexts are $O(B^D)$. In explored regions with bounded context growth, the transform does not change asymptotic branching; otherwise we rely on guardrails (Appx. F) and make no worst-case claim.

**Certificates emitted.** (i) a *partition certificate* that witnesses child leaf-set disjointness and $N(v) = \sum_{u \in \mathrm{Ch}(v)} N(u)$; (ii) a *finiteness certificate* for local finiteness; (iii) a *stable leaf-ID* policy (leaf IDs are canonical hashes of (state, prefix-context, leaf-spec)) so replay is deterministic.

# I APPENDIX I: TOY 4-LEAF REPLAY

**Toy 4-leaf replay.** Root children $u_1, u_2$ with $N(u_1)=3$, $N(u_2)=1$; leaves under $u_1$: $P_1, P_2, P_3$; under $u_2$: $P_4$.

*Logged uniforms.* $U_r=0.20$ (race at root), $W_r=0.70$ (winner selection at root), $U_{u_2}=0.37$ (residual for $u_2$). Quantile winner: since $W_r < 3/4$, $I_r=u_1$.

Then $t(r)=-\log(1-0.20)/4 \approx 0.055786$, so $-\log t(r) \approx 2.884$. Winner reuse (Exact): $t(u_1)=t(r)$. For $u_2$, $E_{u_2}= -\log(1 - 0.37) \approx 0.462$, hence $t(u_2) \approx 0.518$ and $-\log t(u_2) \approx 0.657$. With $M_\tau(r)=5$, $M_\tau(u_1)=4.5$, $M_\tau(u_2)=4.2$:

$$\mathrm{Key}(r)\approx 5 - \log t(r) \approx 7.886, \quad \mathrm{Key}(u_1)\approx 4.5 - \log t(r) \approx 7.386, \quad \mathrm{Key}(u_2)\approx 4.2 - \log t(u_2) \approx 4.859.$$

*Surrogate mode.* With $N_{\mathrm{ub}}(r){=}6$ and the same $U_r$, $\hat{t}(r){\approx}0.03719$ so $-\log \hat{t}(r){\approx}3.288$, making $\widehat{\mathrm{Key}}(r){\approx}8.288$ (conservative). Children would be pushed with parent-anchored provisional keys $M_\tau(u) - \log \hat{t}(r)$ until popped.

## J    APPENDIX J: ADDITIONAL SYSTEM FIGURES

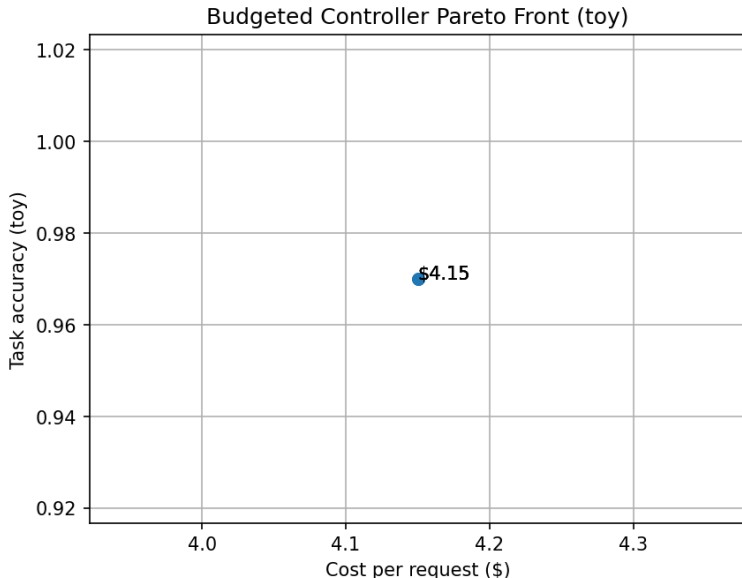

Figure 3: Budgeted controller Pareto front (toy). Operating points under price/SLO; a strict SLO yields a single point.

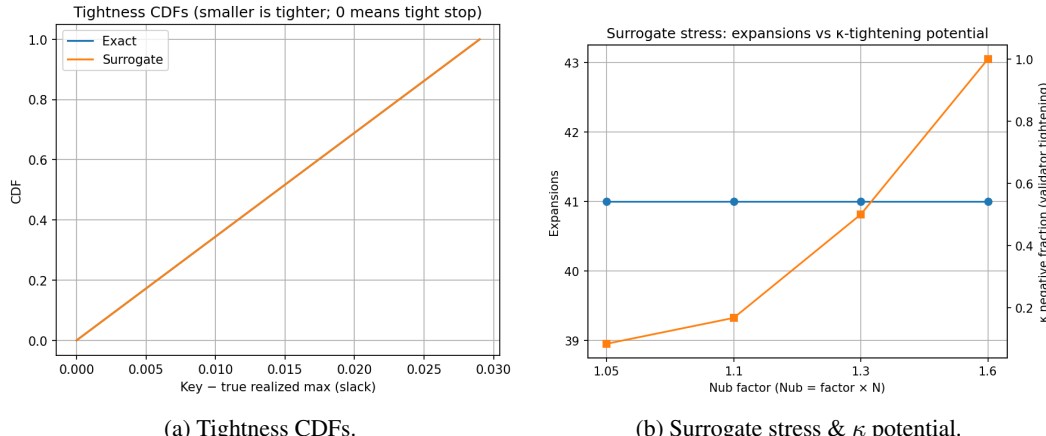

(a) Tightness CDFs.  (b) Surrogate stress & $\kappa$ potential.

Figure 4: **Key tightness and surrogate behavior.** (a) CDF of $\mathrm{Key} - \max_{\mathrm{subtree}}(\mathrm{realized})$ shows **Exact** is tight at stop (slack $\approx 0$) and **Surrogate** is close because parent-anchored keys remain informative. (b) Increasing $N_{\mathrm{ub}}$ makes keys more conservative (more expansions) but yields more validator tightening (higher fraction of negative $\kappa = \log(N/N_{\mathrm{ub}})$); logs show mean $\kappa \approx -0.257$.

## K  APPENDIX K: SCALING AND TRUNCATION DIAGNOSTICS

### K.1  TRUNCATION IN SCORE UNITS AND AN ABSOLUTE LOGSUMEXP BOUND

**Definition of $B_k$.**  For a given prefix node, let $B_k$ be an *upper bound on the number of admissible leaves at depth $k$* under the grammar and compilation constraints (per-depth leaf counts after context indexing; not branching factors).

**Absolute tail bound (no denominator).**  Let $c_{\min}^s > 0$ be a per-step *decrease* in the best possible score, and let $s_{\mathrm{ref}}$ be any certified upper bound on the score of any complete path under the current

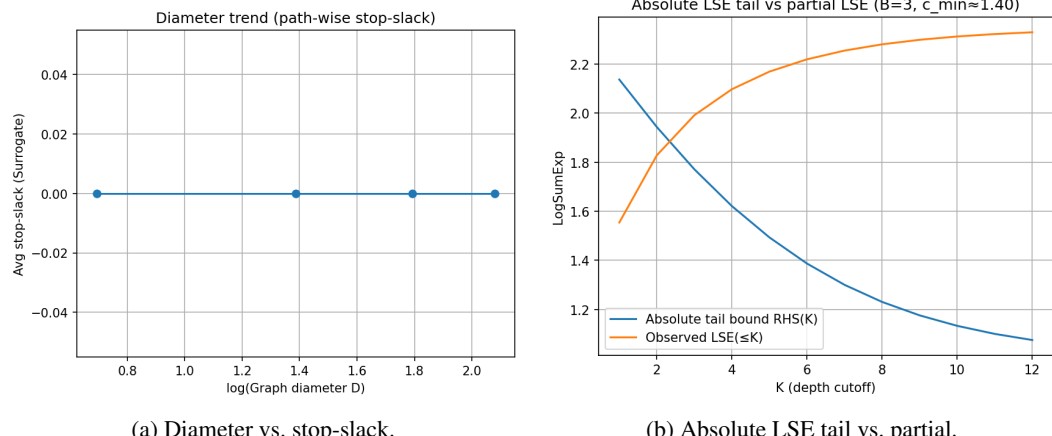

(a) Diameter vs. stop-slack.  (b) Absolute LSE tail vs. partial.

Figure 5: **Tight stopping and LSE truncation certificate.** *Left:* Average stop–slack $\max\{0, \max_{v \in \mathcal{F}} \mathrm{Key}(v) - B^*\}$ (score units) vs. $\log D$, where $D$ is the graph diameter (maximum root-to-leaf steps); values stay $\approx 0$, indicating tight stopping without runaway path cost as graphs grow. *Right:* Absolute LogSumExp decomposition (Eq. equation 1): the certificate uses the maximum between the partial term $\mathrm{LSE}(\leq K)$ and the tail term $s_{\mathrm{ref}} + \log \sum_{k>K} B_k e^{-kc_{\min}^s}$ (up to the $+\log 2$ constant). When $Be^{-c_{\min}^s} < 1$ the tail decays geometrically; after small $K$, the partial term dominates, certifying safe truncation in score units.

prefix (e.g., $s_{\mathrm{ref}} = M_\tau(\mathrm{root})$). Then the overall $\mathrm{LSE} = \log \sum_P e^{s_{\det}(P)}$ obeys

$$\mathrm{LSE(all)} \;\leq\; \max\Big(\mathrm{LSE}(\leq K),\; s_{\mathrm{ref}} + \log\Big(\sum_{k>K} B_k\, e^{-k\, c_{\min}^s}\Big)\Big) \;+\; \log 2. \tag{1}$$

*Proof.* For any $P$ at depth $k$, $s_{\det}(P) \leq s_{\mathrm{ref}} - kc_{\min}^s$. Hence $\sum_{\mathrm{depth}(k)} e^{s_{\det}(P)} \leq B_k\, e^{s_{\mathrm{ref}} - kc_{\min}^s}$. Summing $k > K$ bounds the tail. Split $\sum e^{s(P)}$ into $\leq K$ vs. $> K$ and apply $\log(x+y) \leq \max(\log x, \log y) + \log 2$. *Mental model:* if $B_k \leq B^k$, the RHS tail becomes geometric with ratio $Be^{-c_{\min}^s}$.

## L    APPENDIX L: EXPERIMENTAL SETUP & REPRODUCIBILITY

We provide run seeds, per-run ledgers (NDJSON), and replay scripts for Mac/commodity hardware. The validator replays keys, stop rules, $\kappa$-tightening, and budget events deterministically using the fixed-point formats in Appx. G.1; figures report means over seeds with identical logged uniforms and PRF domain/salt.

## M    APPENDIX M: FULL PSEUDOCODE

---

**Algorithm 1 Best-First PaM (Exact / Surrogate) with Fallback (NoCert)**

---

1: **Input:** grammar $\mathcal{G}$, mode $\in \{\texttt{Exact}, \texttt{Surrogate}, \texttt{Fallback}\}$, admissible $M_\tau$, bounds $(\{B_k\}, c^s_{\min}, K)$

2: Initialize frontier PQ $\mathcal{F}$ (key tie-breaker: lexicographic public IDs); leaf PQ $\mathcal{L}$ (Fallback only); $B^* \leftarrow -\infty$

3: **if** mode$==$ $\texttt{Exact}$ **then**

4:     compute $N(\text{root})$; draw $U(\text{root}) \in (0,1)$; set $t(\text{root}) = -\log(1 - U(\text{root}))/N(\text{root})$

5:     push root with $\text{Key}(\text{root}) = M_\tau(\text{root}) - \log t(\text{root})$; log $(U(\text{root}), N(\text{root}))$

6: **else if** mode$==$ $\texttt{Surrogate}$ **then**

7:     get $N_{\text{ub}}(\text{root})$; draw $U(\text{root}) \in (0,1)$; set $\hat{t}(\text{root}) = -\log(1 - U(\text{root}))/N_{\text{ub}}(\text{root})$

8:     push root with $\widehat{\text{Key}}(\text{root}) = M_\tau(\text{root}) - \log \hat{t}(\text{root})$; log $(U(\text{root}), N_{\text{ub}}(\text{root}))$

9: **else**                                                        ▷ Fallback (NoCert)

10:     push root with $M_\tau(\text{root})$; initialize $\mathcal{L} \leftarrow \emptyset$; mark $\texttt{claim\_type=NoCert}$

11: **while** true **do**

12:     **(optional)** BUDGETCONTROLLER updates $(\varepsilon_{\text{used}}, \text{price}_{\text{spent}}, \text{latency}_{\text{acc}})$ and logs tallies

13:     **if** any budget exhausted **then**

14:         mode $\leftarrow$ $\texttt{Fallback}$; log BUDGETFAIL

15:     **if** mode$==$ $\texttt{Exact}$ **and** $\max_{v \in \mathcal{F}} \text{Key}(v) \le B^*$ **then**

16:         **stop** with run-wise certificate

17:     **if** mode$==$ $\texttt{Surrogate}$ **and** $\max_{v \in \mathcal{F}} \widehat{\text{Key}}(v) \le B^*$ **then**

18:         **stop** with run-wise certificate

19:     $v \leftarrow \arg\max \mathcal{F}$                                    ▷ pop

20:     **if** $v$ is leaf $P$ **then**

21:         **if** mode$==$Exact **then**

22:             $E \leftarrow t(P)$                    ▷ leaf arrival from the realized race; no new randomness

23:             $U_P \leftarrow 1 - e^{-E}$                              ▷ logged for replay

24:         **else**                                                    ▷ Surrogate

25:             draw or PRF-derive $U_P \in (0,1)$;   $E \leftarrow -\log(1 - U_P)$

26:         $V \leftarrow \underbrace{(-C(P))}_{s_{\det}(P)} - \log E$

27:         $B^* \leftarrow \max(B^*, V)$;   LOG $U_P$;   **continue**

28:     **if** mode$==$ $\texttt{Exact}$ **then**

29:         draw $W(v) \in (0,1)$; set $I_v \leftarrow$ QUANTILECAT$\big(W(v); \{N(u)/N(v)\}_{u \in \text{Ch}(v)}\big)$; log $W(v)$

30:         **for** child $u \in \text{Ch}(v)$ **do**

31:             **if** $u == I_v$ **then**

32:                 $t(u) \leftarrow t(v)$

33:             **else**

34:                 draw $E_u \sim \text{Exp}(N(u))$; $t(u) \leftarrow t(v) + E_u$

35:             push $u$ with $\text{Key}(u) = M_\tau(u) - \log t(u)$; log any uniforms used

36:     **else if** mode$==$ $\texttt{Surrogate}$ **then**

37:         draw $U(v)$; set $\hat{t}(v) = -\log(1 - U(v))/N_{\text{ub}}(v)$; log $(U(v), N_{\text{ub}}(v))$

38:         **for** child $u \in \text{Ch}(v)$ **do**

39:             push $u$ with provisional $\widehat{\text{Key}}(u) = M_\tau(u) - \log \hat{t}(v)$

40:     **else**                                                        ▷ Fallback

41:         **for** child $u \in \text{Ch}(v)$ **do**

42:             **if** $u$ is leaf $P$ **then**

43:                 $U_P \leftarrow \text{PRF}(\text{id}(P))$; $G \leftarrow -\log(-\log U_P)$; score $\leftarrow G/\tau - C(P)$; push $P$ into $\mathcal{L}$

44:             **else**

45:                 push $u$ into $\mathcal{F}$ with bound $M_\tau(u)$

46:         **if** $\mathcal{L} \ne \emptyset$ **and** $\max_{P \in \mathcal{L}} \text{score}(P)$ dominates tracked bounds **then**

47:             **stop** (NoCert; heuristic)

---

## N  APPENDIX N: BUDGETED CONTROLLER DETAILS

We equip the router with per-request budgets for privacy, price, and latency while preserving the two-mode frontier invariants of Secs. 4–5. Let the *key slack* at a frontier node $v$ be

$$\text{slack}(v) = \begin{cases} \text{Key}(v) - B^* & \text{(Exact mode),} \\ \widehat{\text{Key}}(v) - B^* & \text{(Surrogate mode).} \end{cases}$$

Smaller slack means we are closer to stopping; thus the BUDGETCONTROLLER may (i) upgrade tools/models/adapters, or (ii) tighten bounds, *provided $M_\tau$ remains admissible and race-independent* (replacing $M_\tau$ by a tighter envelope is allowed) and decisions depend only on DP outputs and public data (post-processing). These actions do not alter the two-mode frontier invariants or the mode-specific stopping rules; all budget events are logged in the ledger.

**Controller interface.**  Each request is initialized with budgets $(\varepsilon_{\max}, \delta)$ for privacy (RDP composed then converted to $(\varepsilon, \delta)$), a monetary cap $\text{price}_{\max}$, and a latency SLO $\text{SLO}_{\text{ms}}$. The router maintains $(\varepsilon_{\text{used}}, \text{price}_{\text{spent}}, \text{latency}_{\text{acc}})$ and logs them (Sec. 10). We enforce the SLO using 95th-percentile latency estimates ("P95") with a safety factor (e.g., $1.2\times$); violations log BUDGETFAIL and trigger downgrade (to cheaper tools/models or `Fallback` if needed).

**RDP accountant (upstream LDP only).**  Upstream LDP emits atoms $(\alpha_i, \varepsilon_i(\alpha_i))$ (stub and hop noises). The router composes them online via Rényi DP, $\varepsilon_{\text{tot}}(\alpha) = \sum_i \varepsilon_i(\alpha)$, and converts to $(\varepsilon, \delta)$ at validation time (Mironov, 2017; Dwork & Roth, 2014). Routing randomness and keys are post-processing of LDP outputs and public grammar (Sec. 11); therefore they do not consume privacy budget.

**Proposition 2** (Budget safety under post-processing). *Let $\mathcal{M}$ be the upstream LDP mechanism and $\mathcal{R}$ the router whose randomness/decisions are measurable w.r.t. $\mathcal{M}(X)$ and public artifacts. Then for every order $\alpha > 1$,*

$$D_\alpha\big(\mathcal{R} \circ \mathcal{M}(X) \,\big\|\, \mathcal{R} \circ \mathcal{M}(X')\big) \leq D_\alpha\big(\mathcal{M}(X) \,\big\|\, \mathcal{M}(X')\big),$$

*so $\mathcal{R} \circ \mathcal{M}$ satisfies the same RDP curve as $\mathcal{M}$ and does not increase $(\varepsilon, \delta)$ at any target $\delta \in (0, 1)$.*

*Proof sketch.* RDP is closed under post-processing (Dwork & Roth, 2014; Mironov, 2017). Given the measurability condition, the router's randomness (uniforms, tie tokens) and decisions are functions of $\mathcal{M}(X)$ and public inputs, hence cannot increase privacy loss. *Note:* if the controller adds any *new* DP mechanism, its atoms must be included in $\varepsilon_{\text{tot}}(\alpha)$.

**Budget-aware model/hop selection (multi-LLM).**  At an expansion of $v$, choose a tool/model $m \in \mathcal{M}$ with attributes $\{\text{price}_m, \text{latency}_m, \varepsilon_m\}$ and an estimated *key-slack reduction* $\widehat{\Delta \text{Key}}(v, m) \geq 0$. Pick

$$m^* \in \arg\max_{m \in \mathcal{M}} \frac{\widehat{\Delta \text{Key}}(v, m)}{\alpha \, \text{price}_m + \beta \, \text{latency}_m + \gamma \, \varepsilon_m} \quad \text{s.t.} \quad \begin{array}{l} \varepsilon_{\text{used}} + \varepsilon_m \leq \varepsilon_{\max}, \\ \text{price}_{\text{spent}} + \text{price}_m \leq \text{price}_{\max}, \\ \text{latency}_{\text{acc}} + \text{latency}_m \leq \text{SLO}_{\text{ms}}, \end{array} \quad (2)$$

with tradeoff weights $\alpha, \beta > 0$ and $\gamma \geq 0$ (to put the denominator on a common scale). Set $\gamma = 0$ when downstream hops add *no* additional LDP noise (i.e., $\varepsilon_m = 0$ unless a hop injects new privatization). Model/tool selection itself is post-processing and free, provided $M_\tau$ remains admissible and race-independent (tightening $M_\tau$ is allowed). This controller is orthogonal to tool-use methods but complementary in practice (Schick et al., 2023; Yao et al., 2023; Patil et al., 2024). When adapters or fine-tuning are employed, recent DP results guide budget constraints (Yu et al., 2022; Charles et al., 2024; Chua et al., 2024).

**Calibration.**  Optionally wrap $\widehat{\Delta \text{Key}}$ with a conformal upper bound to get a conservative gain $\widehat{\Delta \text{Key}}^\uparrow$; this affects only selection, not key validity. The ledger logs predicted vs. realized slack changes as `dkey_pred` and `dkey_real`.

**Soundness w.r.t. keys and stop rules.** Budgeted selection does not modify the race $\{t(\cdot), \hat{t}(\cdot)\}$ nor the admissibility of $M_\tau$; it only changes which actions we execute to *compute/tighten* deterministic bounds. Hence the frontier coverage results and Theorem 2 remain valid.

**Proposition 3** (Frontier soundness under budgeted selection). *Suppose $M_\tau$ remains admissible and race-independent, and the controller obeys the constraints in equation 2. Then (i) the frontier invariant of Lemma 3 (and Theorem 1 in Exact mode) holds unchanged; (ii) when budgets are exhausted and we downgrade to* NoCert, *the run remains replayable and the ledger/validator protocol remains correct. Moreover, if $M_\tau$ depends only on grammar/prefix constraints (not on stochastic tool outputs), it remains admissible under any model choice.*

*Proof idea.* Selection changes neither $t(\cdot)$ nor the parent anchoring of $\hat{t}(\cdot)$; keys are still computed as in Sec. 5. Downgrade alters only the claim type and stop rule (Algorithm 1, Sec. 9); replayability follows from logging uniforms/budgets/tie tokens.

**Validator responsibilities.** Validators recompute the RDP composition from logged atoms and re-derive $(\varepsilon, \delta)$; confirm that the inequalities in equation 2 hold at every step; verify any BUDGETFAIL-induced downgrade; and check that any applied $\kappa$ comes from public counts only. Because budgets and keys are logged in fixed point (Sec. 10), these checks are deterministic.

## O   APPENDIX O: REAL TOOL-USE PIPELINE (SMALL EVAL)

We compile a retrieval→summarize→calculator pipeline into a context-indexed prefix–DAG whose children partition descendant leaves; the search runs in Exact/Surrogate/Fallback modes with a run-wise ledger. We evaluate $n=20$ queries on commodity Mac hardware. Table 3 reports expansions, wall-clock, stop-slack at termination, ledger replay, and surrogate $\kappa$-tightening (from public counts). Repro: `make real-pipeline` (writes `appendix_table_real_pipeline.csv`).

---

[3]Fallback is NoCert; we report 0 for comparability, but stop-slack is not a run-wise guarantee in this mode.

| Mode | Expansions | Wall (s) | Stop-slack | Ledger | Exp/q (±95%) | Wall/q ms (±95%) | $\kappa$-tighten (mean) |
|---|---|---|---|---|---|---|---|
| Exact | 124 | 0.023 | 0.000 | pass | $6.20 \pm 1.12$ | $1.16 \pm 0.64$ | n/a |
| Surrogate | 232 | 0.055 | 0.000 | pass | $11.60 \pm 0.81$ | $2.75 \pm 2.34$ | yes (232/232), $\overline{\kappa} = -0.549$ |
| Fallback | 332 | 0.024 | $0.000^3$ | pass | $16.60 \pm 0.81$ | $1.20 \pm 0.20$ | n/a |

Table 3: **Real tool-use micro-experiment (retrieval→summarize→calculator),** $n{=}20$ **queries.** We compare three routing modes: **Exact** (run-wise certificate with exact counts), **Surrogate** (safe upper-bound counts; keys conservative but validator tightens with $\kappa = \log(N/N_{\mathrm{ub}})$), and **Fallback** (no run-wise certificate). *Expansions* (lower is better) and *Wall* report total cost; per-query means $\pm 95\%$ CIs are shown; *Stop-slack* $= 0$ indicates a tight stop ( $\max_{v \in \mathcal{F}} \mathrm{key}(v) \leq B^*$ ); *Ledger* shows deterministic replay; $\kappa$-*tighten* reports nodes tightened and mean $\kappa$ (more negative is tighter). Exact expands least; Surrogate expands more but is ex-post tightened everywhere; Fallback explores the most.

## P  APPENDIX P: DP-TRAINED ADAPTERS (TRANSPARENCY TABLE)

We list the adapters considered by the controller, their training DP certificates $(\varepsilon_{\mathrm{train}}, \delta_{\mathrm{train}})$, and how often they were selected per mode. We also report the mean *inference* $\varepsilon$ used, which remains zero under our post-processing posture. The CSV is generated by `make adapter-table` (file: `appendix_table_adapters.csv`) and can be re-created on any Mac with our repo.

Table 4: **DP-trained adapters and whether the controller used them (transparency).** Each row lists an adapter considered by the router, its DP training certificate ($\varepsilon_{\text{train}}, \delta_{\text{train}}$), how many times the budget-aware controller selected it in each routing mode (Exact/Surrogate/Fallback) and in total, and the average per-hop inference privacy $\varepsilon$ actually consumed. Under our post-processing posture, inference $\varepsilon=0$ indicates no additional DP noise at inference; training $\varepsilon$ is incurred offline. Counts are aggregated over our reproducibility runs and will vary with seeds.

| Adapter | Tier | DP cert | $\varepsilon_{\text{train}}$ | $\delta_{\text{train}}$ | Chosen (runs) | | | | Inference $\varepsilon$ (avg) |
|---------|------|---------|------|------|-------|-----------|----------|-------|-------------|
| | | | | | Exact | Surrogate | Fallback | Total | |
| LoRA-A-small | Small | certA | 2.0 | $1 \times 10^{-6}$ | 240 | 0 | 0 | 240 | 0.0 |
| LoRA-B-medium | Medium | certB | 3.5 | $1 \times 10^{-6}$ | 0 | 680 | 0 | 680 | 0.0 |
| LoRA-C-large | Large | certC | 6.0 | $1 \times 10^{-6}$ | 0 | 0 | 0 | 0 | 0.0 |
| (no-adapter) | None | — | — | — | 0 | 0 | 402 | 402 | 0.0 |

