# OpenReview forum: "Auditable Early Stopping for Agentic Routing: Ledger-Verified Run-Wise Certificates under Local DP"
_ICLR.cc/2026/Conference — Submitted to ICLR 2026_

### Official Review · Reviewer_LcDD · 2025-10-30

**Soundness:** 1
**Presentation:** 1
**Contribution:** 1
**Rating:** 2
**Confidence:** 2

**Summary:**

This paper appears to present something, but it is not entirely clear what. There is no real introduction, no motivation for the work, and little to go on for meaning unless the reader happens to have worked on exactly the same thing and uses the same terminology. Since tghe paper is written only for a very specialized small audience that is deeply enmeshed in everything described, this is not suitable for conference review in it's present form. If other reviewers feel differently, that will surface through the review process and the paper can be evaluated on its merits.

**Strengths:**

This paper appears to be written in a precise mathematical style.

**Weaknesses:**

Not understandable to a general audience.

**Questions:**

What is this about? In particular, many of the references are to work that predates agentic AI, so it is not clear whether technical terms used here refer to modern agentic systems or to some older computation model or algorithms.

---

### Official Review · Reviewer_cBMf · 2025-11-01

**Soundness:** 1
**Presentation:** 1
**Contribution:** 2
**Rating:** 0
**Confidence:** 2

**Summary:**

The paper presents a way to use local differential privacy in best-first routing with early stopping and certification. The paper also aims to create a repayable ledger or log.

**Strengths:**

The paper wants to create an auditable system where the routing decisions both satisfy differential private and are auditable. The use of post-processing property of DP seems like the right approach. The evaluation with respect to different baselines shows that the method outperforms them.

**Weaknesses:**

The paper is incomprehensible to this reviewer in its present form. It begins without defining a problem statement and instead presents a very specific running example with undefined terms. There is no context given for the problem that a general reader can understand.

The technical claims are not possible to check. There are so many terms used which have no definitions before they are used (e.g. certificates, context-indexed, PRF, etc.). Citations to the prior algorithms are omitted so one cannot follow them to fill in the gaps. Barring these syntactic issues, the deeper concern is that there are formal claims throughout which lack a proof. For example, one page 3 Lemma 1 doesn't have a proof, it has a sketch, and Lemma 2 has merely a justification.

Given the state of the manuscript, a technical evaluation or acceptance of the presented ideas is very difficult.

**Questions:**

None.

---

### Official Review · Reviewer_dyL6 · 2025-11-08

**Soundness:** 2
**Presentation:** 1
**Contribution:** 1
**Rating:** 0
**Confidence:** 2

**Summary:**

The paper is very difficult to read. I could not go beyond the first few pages.

The paper does not clearly explain the motivation behind the problem it aims to solve. Many technical terms are introduced without adequate explanation. A few examples are listed below:

Line 013: The term “local” in local differential privacy is not defined.

Line 015: The term “leaf” is used without a prior definition of a DAG. Why does a node contain a key?

Line 019: Exact mode and Surrogate mode are mentioned with little explanation of their meaning.

Line 024: The analysis refers to DAGs, but the experiments are conducted on graphs—this inconsistency is unclear.

Line 032: The paper mentions a “grammar” without introducing its purpose or explaining how it relates to local differential privacy.

I may be an outsider to this specific field, but a well-written paper should still make its goals and motivations clear without requiring additional background reading.

**Strengths:**

I could not finish reading the paper, so I am unable to assess its strengths.

**Weaknesses:**

The paper is difficult to read for people outside the field.

**Questions:**

What is the motivation behind your work?

---

### Official Review · Reviewer_q8hu · 2025-11-10

**Soundness:** 3
**Presentation:** 2
**Contribution:** 3
**Rating:** 4
**Confidence:** 3

**Summary:**

The authors explore the easy stopping for best-first routing agents under local differential privacy (LDP) and develop a run-wise certificate mechanism by coupling node pruning keys to the same exponential races. The discussed issue is significant, and the authors conduct several experiments to verify the effectiveness of the work.

**Strengths:**

1. The discussed issue in terms of privacy and auditability is significant.
2. Several experiments are performed.

**Weaknesses:**

1. The presentation of the work should be improved. This paper does not fully illustrate the research background and motivations.
2. The work lacks a formal and detailed definition of the studied problem;
3. Some symbols are not predefined before being used. For example, in the first paragraph of the Introduction, $s_{det}$, $C(P)$ are not explained.
4. A literature review is not provided. Are there any similar latest works on the same topic?
5. The comparison seems a little insufficient.

**Questions:**

Refer to the weakness.

---

### Official Review · Reviewer_3owF · 2025-11-17

**Soundness:** 2
**Presentation:** 1
**Contribution:** 1
**Rating:** 2
**Confidence:** 4

**Summary:**

The paper proposes a method for applying local differential privacy to best-first routing with early stopping and certification, and further seeks to construct a verifiable, replayable ledger or log.

**Strengths:**

This paper addresses an important problem: ensuring that the auditable system satisfies differential privacy and allows routing decisions to be audited. They have conducted experiments against selected baselines to demonstrate how their method has improved upon them and to justify their claims.

**Weaknesses:**

- The issue from the paper is that it is very difficult to decipher the motivation behind the work, and the concepts discussed are not introduced in a logical way.
- The formal definition of the problem is missing, and most of their definitions are lacking mathematical proof. This is a dangerous trend for the paper, making me hard to trust the paper and its claims.
- Given the current state of the manuscript, a thorough technical assessment or validation of the ideas is challenging.

**Questions:**

1. For "Run-wise certificate" the statement itself does not make sense. Can you please expand upon it?

---

### Meta-Review · Area_Chair_xypn · 2026-01-09

**Summary:**

Reviewers agree the paper addresses an important problem—applying local differential privacy to best-first routing with early stopping and auditability/certification—and they appreciate that experiments were conducted and show improvements over selected baselines. However, the dominant concern across the reviews is that the paper is very difficult to understand, with unclear motivation, missing formal problem definition, and many terms/symbols introduced without explanation. Several reviewers report they could not meaningfully assess correctness because key technical claims lack complete proofs (or are presented only as sketches/justifications), and the manuscript lacks sufficient background, related work positioning, and comparisons. Given these issues, the committee concluded the submission is not yet at the level required for acceptance.

**Reviewer Concerns:**

Based on the reviews, some concerns (e.g., clarifying terminology such as “run-wise certificate,” explaining inconsistencies like DAG terminology vs graph experiments, or adding missing context and related work) could potentially be partially addressed by a rebuttal. However, the most serious weaknesses appear structural and likely remain outstanding: the absence of a clear formal problem statement, insufficiently developed proofs for core claims, and overall readability issues severe enough that multiple reviewers could not complete or verify the work. These unresolved issues substantially limit confidence in the correctness and evaluability of the contribution and ultimately support the reject decision.

**Reviewer Scores:**

If all reviewers had fully participated in discussion, the two strong-reject reviewers (dyL6, cBMf) would likely remain strongly negative (0 → 0, at most 0 → 1) because their primary issue was incomprehensibility and inability to validate claims. The reject reviewers (LcDD, 3owF) might at most shift slightly upward (2 → 3) if the rebuttal and discussion clarified the paper’s motivation and definitions, but would likely stay at reject due to missing formalism and proof gaps. The most positive reviewer (q8hu, score 4) could plausibly increase to a borderline accept (4 → 5) if presentation, definitions, and comparisons were convincingly strengthened, though the overall decision remains rejection given the breadth and severity of unresolved concerns.

---

### Decision · Program_Chairs · 2026-01-26

Reject